# The Life cycle Assessment Integrated with the Lexicographic Method for the Multi-Objective Optimization of Community-Based Rainwater Utilization

**DOI:** 10.3390/ijerph20032183

**Published:** 2023-01-25

**Authors:** Yi Li, Wenjun Xu, Wenlong Zhang, Youyi Huang, Fenfen Wan, Wei Xiong

**Affiliations:** 1Key Laboratory of Integrated Regulation and Resource Development on Shallow Lakes, Ministry of Education, College of Environment, Hohai University, Nanjing 210098, China; 2School of Architecture and Civil Engineering, Xiamen University, Xiamen 361000, China; 3School of Hydraulic and Environmental Engineering, Changsha University of Science & Technology, Changsha 410114, China

**Keywords:** community-based rainwater utilization, life cycle assessment, multi-objective optimization, lexicographic method, uncertainty

## Abstract

Community-based rainwater utilization (CB-RWU) has the advantage of easy maintenance and multiple benefits. However, its promotion proves to be a complicated task due to difficulties in quantifying and evaluating external benefits. This study integrated the life cycle assessment (LCA) with a multi-objective optimization model to optimize the relationship among all stages of CB-RWU, considering the trade-offs among the benefit–cost ratio, water-saving efficiency and environmental impact. The LCA results identified abiotic depletion potential for fossil fuels (ADPF) as the key impact indicators throughout the life cycle of CB-RWU. The optimal solution from the lexicographic method was 0.3098, 28.47% and 24.68 MJ for the benefit–cost ratio, water-saving efficiency and ADPF, respectively. Compared with the traditional optimization method, the lexicographic method improved the three object functions by 26%, 43% and 14%, respectively. The uncertainty of the environmental impact was the highest (C_V_ = 0.633) with variations in the floor area ratio, total runoff coefficient and reservoir volume. Changes in the total runoff coefficient were the main source of the uncertainty, which suggested that more attention should be paid to the area ratio of each underlying surface. In addition, economic support from the government is urgently required for the further promotion and development of CB-RWU.

## 1. Introduction

Rapid industrialization and urbanization have enhanced environmental problems, including water scarcity and pollution, and have thus become a bottleneck for the sustainable development of the economy and society [1,2,3]. A variety of measures have been taken to alleviate water scarcity and pollution to promote the harmonious coexistence of human society and natural systems. As an important part of unconventional water source utilization and the construction of sponge cities, urban rainwater utilization can maximize the on-site utilization of rainfall. It usually involves multiple stages, including rainfall, harvesting, storage, treatment, distribution and rainwater use. The small-scale (e.g., individual household-based) roof rainwater harvesting (RWH) system is the most common form of urban rainwater utilization and is adopted widely in China and other parts of the world [4,5,6,7]. However, the poor operation and maintenance of treatment systems inhibit these small-scale units to operate efficiently and yield desirable results [8]. To overcome this limitation, community-based rainwater utilization (CB-RWU), also known as the ‘centralized’ or ‘neighborhood’ rainwater utilization, can be implemented, with multiple external benefits such as alleviating the contradiction between urban water supply and demand as a supplementary water source, effectively reducing runoff pollution and alleviating waterlogging [9,10,11,12,13,14,15]. The policy for rainwater development and utilization is a crucial means to convert the aforementioned external benefits into investment benefits and promote their further development. In particular, the comprehensive evaluation and optimization of CB-RWU integrates the environmental, economic and social benefits for policy formulation. However, determining how to balance the trade-offs among multiple benefits under the uncertainties during all stages, including rainfall, harvesting, storage, treatment, distribution and rainwater use, remains a key issue during the evaluation and optimization process.

Several studies have been developed to evaluate the economic and environmental benefits of urban rainwater utilization. However, most studies only focus on ‘small-scale’ RWH systems and evaluate these benefits under different technologies [16,17], users [18,19,20] and scenarios [21,22,23,24,25]. Studies on CB-RWU are limited and generally evaluate the corresponding economic and environmental benefits. Sun et al. quantitatively calculated the carbon sequestration, oxygen reduction and the cost–benefit ratio of CB-RWU in six sponge communities using the water balance method, an alternative engineering method and the market value method [26]. Gu et al., constructed an evaluation index system for the control potentiality of runoff non-point source pollution in urban residential communities in Shanghai. However, the requirements of the sustainable utilization of water resources can no longer be met by economic-oriented assessment methods [27]. Furthermore, traditional environmental impact assessment methods only focus on the operation and maintenance (O&M) phase of a system, leading to difficulties in quantifying the impact of CB-RWU on the environment and human health throughout the whole life cycle.

Life cycle assessment (LCA) can provide a quantitative environmental impacts analysis using a cradle-to-grave approach and has been widely used to evaluate the environmental impacts of rainwater utilization [28,29]. By Taking five RWH systems as the research object, Farago et al., quantified the potential environmental impacts using LCA and assessed the economic system value in terms of total value added [21]. Angrill et al. applied LCA to quantify the material consumption and environmental impacts of eight RWH scenarios with different city densities and storage tank locations in residential areas, accounting for several stages (e.g., rainwater harvesting, storage and distribution) [30]. Spatari et al. considered the rainwater harvesting and treatment stages to compare the energy consumption and greenhouse gas (GHG) emissions within the whole life cycle between cities with and without low-impact development (LID) facilities [31]. However, the trade-offs among different stages and the uncertainty in the changes of influencing parameters are not accounted for in the majority of the previous studies despite their importance on the accurate assessment of the comprehensive benefits and optimization of the CB-RWU. In this case, a more systematic optimization of CB-RWU from the perspective of comprehensive benefits is required.

Multi-objective optimization can simultaneously optimize multiple objective functions to obtain the optimal solution satisfying the decision preference. Numerous studies have been developed to optimize rainwater pipe networks or LID facilities using different objective functions [32,33] and optimization methods [34,35,36,37]. Liu et al., developed a decision-support tool to determine the best LID facility plan in the study area [38]. Baek et al. proposed a feasible method to plan the location and scale of various LID facilities [39]. In terms of CB-RWU, the optimal size of a RWH tank and rooftop area have been determined with total cost and water saving as the objective functions in a study in Malaysia [9]. However, few studies consider the differences in the importance of objective functions. As an integral component to the promotion of CB-RWU, the economic benefit is the most significant among the three objective function types [40]. CB-RWU will not be effectively implemented without economic incentive. Thus, any water resources or environmental benefits are of no practical value without first demonstrating the economic benefits. The importance of all objective functions is considered equal in traditional multi-objective optimization methods, which may lead to an optimal solution with high environmental benefits and low economic benefits. Therefore, in order to ensure the reliability of the optimization results, water resources and environmental benefits need to be improved on the premise of ensuring economic benefits. The lexicographic method is employed for a variety of objective functions according to their importance and places different emphases on them when solving multi-objective optimization problems to guarantee the main objective function. The key to this method is to accurately evaluate the value of each objective function. Therefore, this study integrates LCA with the lexicographic method for the multi-objective optimization of CB-RWU, providing a reference for the evaluation and optimization methods of CB-RWU research.

The objectives of this study are to: (1) obtain reliable assessment results of the environmental impacts of CB-RWU based on the LCA of 80 sets of scenarios and identify the main indicator for the environmental impacts in the optimization model; (2) optimize the relationship between all stages of rainwater utilization under the trade-offs among economic, environmental and water resource benefits through the lexicographic method; and (3) clarify the influence of key variables on the environmental, economic and water-resource benefits of CB-RWU and quantify the source of uncertainty for each objective function through a Monte Carlo simulation.

## 2. Methods

### 2.1. Study Area

Dadeng Island in Xiamen, located on the southeast coast of China, was selected as the study area. It has a total area of 41.61 km^2^, with a population of 90,000, including 68,000 residents and 22,000 villagers. Figure 1 presents the location and land use type of Dadeng Island. A reservoir of flood-regulating lakes is around the island, with a total water area of 1.18 km^2^.

Xiamen has a typical East Asian monsoon climate with the average precipitation reaching 1249 mm annually. Autumn and winter occur during October to February and are mainly controlled by the cold and dry air of the mainland; and February to September is spring and summer, which are controlled by the warm and humid air from the ocean. Analysis of daily rainfall data from 1980–2009 reveals that rainfall in this area is generally concentrated from April to August, of which the total rainfall in June, July and August accounts for 43% of the annual rainfall. In general, problems exist in this area, including a highly uneven rainfall distribution within a year, a high population density with large water demand and serious water scarcity. Fortunately, abundant precipitation brings great potential for rainwater utilization. At present, Xiamen adopts the drainage system of rain and pollution diversion and is one of the first batch of pilot cities for sponge city construction. It has promulgated many related standards and policy documents. The rainfall and economic data, as well as some of the additional parameter values, are local data of the Xiamen area in this study, which are detailed in Section 2.3.3.

### 2.2. LCA Model

The LCA modelling was conducted using GaBi Professional software [41], which is widely adopted in other LCA applications [42,43,44]. The industrial ISO 14,044 standard (2006) was followed.

#### 2.2.1. Rainwater Utilization Mode

The most common RWH and utilization system of an ordinary community was taken as the study object (Appendix A). The underlying surface of the catchment area was divided into four types, namely, hard pavement, permeable pavement, roof and green land. The initial rainwater of permeable pavement and green land was used to supplement groundwater after infiltration, while that of hard pavement and roof was filtered by an initial split-flow device to remove large floating objects. The rainwater then entered the underground reservoir to be fully precipitated and was treated to reach the standard of reclaimed water reuse (COD ≤ 30 mg/L, SS ≤ 5 mg/L) by sand filtration and chlorine disinfection. All runoff generated was assumed to be collected for rainwater utilization after the initial split-flow. Based on the total runoff coefficient, reservoir volume and the floor area ratio (FAR), the actual available rainwater for utilization is calculated by Equations (3)–(8) in Section 2.3.1. According to the *Standard for design of building water supply and drainage* (GB50015-2019), the total amount of water used for daily green irrigation and road watering is less than one-tenth of the household water consumption. Therefore, this study assumed that rainwater was only used for other non-potable uses such as toilet flushing and other indoor uses. It was assumed the demand for these uses would be met by traditional drinking water only under the condition of insufficient rainwater. The daily non-potable water demanded is calculated by Equations (9) and (10) in Section 2.3.1.

#### 2.2.2. Goal and Scope

The objective of this study is to quantify the environmental impact of CB-RWU through LCA and provide a foundation for multi-objective optimization research. The system boundary of LCA included the whole process from rainwater to CB-RWU for 30 years, covering the construction, O&M and end disposal phases (Figure 2). The construction phase considered the environmental impacts caused by production, transportation and the on-site construction of all raw materials. The O&M phase considered the replacement material production and chemical consumption required for the system operation. In addition, energy consumption was considered in all phases. The pollution caused by the system included wastewater, waste gas and other forms of pollutants. The functional unit was 1 m^3^ of water supply. All the relevant data, including the durable years, data on rainfall, the consumption of raw materials, chemicals, fossil energy, electric energy, etc., and the discharge of air, water and solid pollutants during the CB-RWU process were collected from the Meteorological Bureau, related standards for rainwater utilization and previous studies [17,45,46,47,48].

#### 2.2.3. Inventory Analysis

The floor area ratio refers to the ratio of the total building area to the net land area of a community, which is closely related to the living comfort of residents. According to the local and national standards, the FAR, total runoff coefficient and reservoir volume ranged between 1–3.1, 0.09–0.42 and 33–500 m^3^, respectively. A total of 80 sets of CB-RWU scenarios with different FAR (1, 1.7, 2.4 and 3.1), total runoff coefficients (0.09, 0.20, 0.31 and 0.42) and reservoir volume (33FAR, 66FAR, 99FAR, 132FAR and 165FAR) were simulated, and a life cycle assessment model was established to analyze the impact of changes in these three factors during the whole life cycle on the environment. Appendix A reports the engineering quantity list.

#### 2.2.4. Life Cycle Impact Assessment

The CML 2001 evaluation model has the advantages of few hypotheses, low model complexity and easy operation. The environmental impact indicators can be divided into three categories, namely, resource depletion, ecosystem and human health. Therefore, this study selected 10 indicators under this evaluation model for the assessment (Table 1). All 10 indicators were initially assessed, while only abiotic depletion potential for fossil fuels was chosen as the representative environmental indicator to become an object function for the multi-objective optimization (Section 3.1).

### 2.3. Multi-Objective Optimization Model

A multi-objective optimization model was established to optimize the relationship among all stages, such as rainwater harvesting, storage and distribution, based on the nondominated sorting genetic algorithm II (NSGA-II) and lexicographic method, considering the differences in the importance of objective functions. The RWH and utilization system of a community were taken as the study object. The optimization objectives were considered from three aspects, including water resource, economic and environmental benefits. The main purpose of CB-RWU was to collect rainwater instead of tap water for various non-potable purposes to relieve the pressure of regional water resources and ensure the water demand of the users. Therefore, water-saving efficiency (WSE) was selected as the main indicator for the water resource benefit. Since the main economic benefit of CB-RWU came from the cost of conventional water resources saved by the use of rainwater, the benefit–cost ratio (BCR) was selected as the main indicator for the economic benefit, considering income and investment comprehensively. According to the results of LCA, the ADPF was selected as the main indicator for the environmental impacts of CB-RWU in the optimization model.

#### 2.3.1. Object Functions

The multi-objective optimization model of CB-RWU systems consisted of three objectives: the maximization of the benefit–cost ratio, the maximization of water-saving efficiency and the minimization of ADPF value for environmental impacts.

The benefit–cost ratio (BCR) of the rainwater utilization system was equal to total revenue divided by total investment:(1)$$\mathrm{BCR}=\frac{\sum_{i=0}^{T}\frac{S_{i}P_{i}}{{\left(1+r\right)}^{i}}}{\sum_{i=0}^{T}\frac{I_{i}+M_{i}}{{\left(1+r\right)}^{i}}}$$
where S_i_ (m^3^) denotes the conventional water resources saved using rainwater in year i; P_i_ (CNY/m^3^) is the price of conventional water resources in year i, which was valued as 3.2 CNY/m^3^ in the study aera [49]; I_i_ (CNY) is the investment of the CB-RWU system in year i (e.g., material, installation, on-site construction and material transportation costs); M_i_ (CNY) is the O&M costs in year i (e.g., electricity, equipment replacement and maintenance costs) [50]; r is the annual interest rate and was valued at 4.06% [51], and T is the total years of operation and was valued at 30 years [52,53]. The electricity price in the study area was 0.667 CNY/kw·h, and other costs were estimated according to the *Investment Estimate Index of Sponge City Construction Project* (ZYA1-02(01)-2018).

The abiotic depletion potential for fossil fuels value (MJ) for the environmental impact is determined using Equation (2). The details about the environmental impacts are described in Section 3.1:(2)$$\mathrm{ADPF}=20.731\mathrm{FAR}\;-\;246.091\psi +0{.036S}_{C\;}+102.996$$
where FAR is the floor area ratio, namely, the ratio of the total building area to the net land area of a community, S_C_ (m^3^) is the volume of the underground reservoir, and Ψ is the total runoff coefficient of the community, which can be expressed as:(3)$$\psi =\frac{1000\sum_{k=1}^{4}I_{t_{k}}}{H_{t}F}$$
(4)$$I_{t_{k}}=\{\begin{array}{c} {0\;,\;H}_{t}\;\leq {\;\delta }_{k}\\ \frac{\psi_{k}\beta_{k}{(H}_{t}{\;-\;\delta }_{k})F}{1000}{,\;H}_{t}>\delta_{k} \end{array}$$
where $I_{t_{k}}$ is the runoff generated on day t by green land (k = 1), permeable pavement (k = 2), hard pavement (k = 3) and roof (k = 4), respectively; Ψ_k_ is the runoff coefficient for each underlying surface, the range of which is from the *technical code for rainwater management and utilization of building and subdistrict* (GB50400-2016) and previous studies [54,55] and shown in Table 2; and β_k_ is the proportion of area for each underlying surface to the total catchment area. According to the *standard for urban residential area planning and design* (GB50180-2018), β_3_ and β_4_ were set as 0.1 and 0.25, respectively, while β_1_ and β_2_ ranged between 0 to 0.65, and δ_k_ is the abandoned flow at the beginning of rainfall. According to the standard (GB50400-2016), δ_1_-δ_3_ was set as 5 mm and δ_4_ as 2 mm; H_t_ is the rainfall on day t and was determined using average daily rainfall data for 30 years (1980–2009); and F (m^2^) is the total area of the catchment area, valued as 30,000 m^2^.

Water-saving efficiency (WSE) is the ratio of the total rainwater used to total water demand and is determined as:(5)$$WSE=\frac{\sum^{\;}Y_{t}}{\sum^{\;}D_{t}}\times 100\%$$
where *Y_t_* (m^3^) is the water used on day *t*, and *D_t_* (m^3^) is the non-potable water demand (e.g., toilet flushing and other indoor uses) on day *t*, calculated using Equations (6)–(9):(6)$$Y_{t}=\{\begin{array}{l} R_{t-1}+I_{t},R_{t-1}+I_{t}-D_{t}\leq 0\\ D_{t},\;\;\;R_{t-1}+I_{t}-D_{t}>0 \end{array}$$
(7)$$R_{t}=\{\begin{array}{l} R_{t-1}+I_{t}-Y_{t},R_{t-1}+I_{t}-S_{c}\leq 0\\ S_{c}-Y_{t},\;\;\;R_{t-1}+I_{t}-S_{c}>0 \end{array}$$
(8)$$R_{t}=\{\begin{array}{l} R_{t-1}+I_{t}-Y_{t},R_{t-1}+I_{t}-S_{c}\leq 0\\ S_{c}-Y_{t},\;\;\;R_{t-1}+I_{t}-S_{c}>0 \end{array}$$
(9)$$I_{t}=\{\begin{array}{l} 0,\;\;\;H_{t}\leq \delta \\ \frac{\psi \left(H_{t}-\delta \right)F}{1000},H_{t}>\delta \end{array}$$
(10)$$D_{t}=\frac{ndN\alpha }{1000}$$
where *R_t_* (m^3^) is the remaining rainfall in the reservoir at the end of day *t*; *n* (L/per^−1^·d^−1^) is the daily water use quota of the study area, taken as 160 L/per^−1^·d^−1^ based on *the norm of industrial water use and living water use* (DB3502/Z 5016-2016); *d* is the number of days which CB-RWU is running, taken as 1 day; α is the proportion of non-potable water in the daily water use quota, which was set as 40% based on *the standard for design of building reclaimed water system* (GB50036-2002); and N is the population of the community and was calculated as:(11)$$N=3.5\;\times \;\frac{\mathrm{FAR}}{25\%}\;\times \;60$$

#### 2.3.2. Constraints

The major constraints for this study are the remaining rainfall in the reservoir and the total catchment area. The initial rainfall in the reservoir was 0 m^3^, and the remaining rainfall was always greater than or equal to 0 m^3^. The volume of the underground reservoir was less than three times the daily demand. The total area of the four underlying surfaces was equal to the total design catchment area, namely, 30,000 m^2^. The FAR of the residential community was between 1 and 3.1.

#### 2.3.3. Datasets

The inventory analysis was conducted using primary and background datasets. The primary data included data on rainfall, the consumption of raw materials, chemicals, fossil energy, electric energy, etc., and the discharge of air, water, and solid pollutants during the CB-RWU process. All primary data were collected from the Meteorological Bureau, related standards for rainwater utilization and previous studies [17,45,46,47,48]. The background data included economic data, such as the price of water, electricity, chemicals and materials, as well as the runoff coefficient, the proportion of area for each underlying surface to total catchment area, the abandoned flow at the beginning of rainfall, etc. All background data were provided by the GaBi database and local and national standards. More specifically, the rainfall and economic data, as well as some of the additional parameter values, were local data of the Xiamen area. When there was no clear provision in local standards, the parameter values were determined with reference to the national standards, which also applied to the Xiamen area. The LCA list of the rainwater utilization system is provided in Appendix A, and is based on Appendix A and the Chinese energy sector data.

#### 2.3.4. Lexicographic Method

As the key to the promotion of CB-RWU, the economic benefit was considered as the most significant among the three types of objective functions in this paper. Without promotion, there would be no water resources and environmental benefits [40]. Furthermore, for CB-RWU, ensuring local water resources benefits and alleviating water shortages were considered more urgent than reducing the global environmental impacts obtained through LCA [56]. Therefore, the priority goal of this optimization model was to maximize the BCR, followed by the improvement of the WSE and the minimization of the ADPF. The NSGA-II and the lexicographic method were adopted to meet these goals (Figure 3). The implementation process follows these pseudocodes:Determine the sequence of objective functions. The first-level goal was to maximum the BCR; the second-level goal was to maximum the WSE; and the third-level goal was to minimize the ADPF.First-level optimization. Solve the optimization model with the maximization of the BCR as the goal under the initial constraints, obtaining a solution set and the maximum BCR.Second-level optimization. Modify the constraints and solve the optimization model with the maximization of the WSE as the goal within the first-level optimal solution range, obtaining a solution set and the maximum WSE.Third-level optimization. Modify the constraints and solve the optimization model with the minimization of the ADPF as the goal within the second-level optimal solution range, obtaining a solution set and the minimum ADPF.Check the solutions of the total runoff coefficient, reservoir volume and FAR. Stop the calculation and output the final optimal solution of the model.

### 2.4. Uncertainty Analysis

#### 2.4.1. Sources of Uncertainty

The CB-RWU system is associated with characteristics, including randomness, ambiguity and uncertainty. The uncertainties originate from the following sources [57]: (1) the randomness and unpredictability of the environment, such as the uncertainty of daily rainfall; (2) the uncertainty of the model structure, resulting in uncertainties in the simulation results; and (3) uncertainty of the data and model parameters, such as missing and distorted data. In this study, we only considered the uncertainty of the model parameters in the process of rainwater harvesting, treatment, storage and distribution, represented by the total runoff coefficient, the reservoir volume and the FAR, respectively. Table 3 presents the values and probability distribution of these three parameters based on the related standards and previous studies [58,59,60].

#### 2.4.2. Monte Carlo Analysis

A Monte Carlo simulation can effectively analyze the uncertainty of complex models based on the probability model or statistical law of random variables [61]. Random sampling was used to extract random parameter values based on the probability distributions of the uncertainty parameters. A total of 1000 sets of uncertainty parameter combinations were obtained and integrated into the objective functions for calculation by a random sampling program. In addition, statistical analysis was carried out on the environmental, economic and water resource benefits of CB-RWU.

## 3. Results and Discussion

### 3.1. LCA Performance Analysis of CB-RWU

Appendix A and Figure 4 present the results of the 10 indicators across all life cycle stages for every scenario. The marine aquatic ecotoxicity potential (MAETP) was observed to have the largest environmental impact. This is related to the concentration of heavy metals discharged into water, which are typically produced in the process of electricity production and coal mine waste treatments in landfills. The second-ranked indicator was the ADPF, which is attributed to the large consumption of fossil energy (e.g., coal, oil and natural gas) during raw material production, material transportation, on-site construction and end disposal. Moreover, according to the Chinese power grid database in GaBi and the National Bureau of Statistics of China, the current power structure in China is still dominated by hard coal power generation, while clean energy only accounts for about 25% of the total power generation. High levels of electricity consumption occurred at all stages of the whole life cycle [62]. This was particularly true during the operation and maintenance stage, where continuous power was required to maintain the operation of water pumps and ensure normal water usage. Therefore, the large use of electricity was also an important reason for the high ADPF.

Further, in terms of the materials, the concrete used in the construction of the tank and the reinforcement, the cement content and the transport of components turned out to be the major contributors to the environmental impacts of concrete construction [63]. In particular, cement clinker was the component with the greatest contribution in the final impact of concrete [64]. For this reason, it was important to select a concrete with a suitable cement content to fulfill the target function. Therefore, coprocessing in the cement industry was recommended to recover energy and materials from waste, thus reducing the environmental impacts of concrete during its life cycle. In terms of the energy consumption, the electricity consumption of the water pump was the main contributor during the rainwater utilization [65]. A review of the energy consumption of the rainwater utilization system showed that the theoretical energy consumption (median 0.20 kWh/m^3^) was much lower than the empirical energy consumption (median 1.40 kWh/m^3^). On the one hand, this result indicated that part of the energy consumption in the system may be ignored in the theoretical analysis; on the other hand, it also indicated a large optimization space of the current rainwater utilization system [66]. Therefore, eliminating or reducing pumping energy was key to reducing the environmental impacts of CB-RWU. For example, an alternate energy mix could minimize impacts of a dominant release contributor.

In general, when the total runoff coefficient of the community was low, the environmental impact of CB-RWU increased simultaneously with the reservoir volume, while the opposite trend was observed when the total runoff coefficient was high. Taking the changes of the FAR into account, further comparisons revealed that the reduction effect of the reservoir volume on the environment was weakened with the increasing FAR under a high runoff coefficient. This is due to the wasted capacity and materials for a large reservoir under a low runoff coefficient, which was amplified following an increase in the daily water demand. In terms of increasing the environmental benefits, communities with a low FAR are more suitable for CB-RWU. The reservoir volume is recommended to be selected with comprehensive consideration of the FAR and runoff coefficients. More specifically, when the total runoff coefficient is low (Ψ < 0.1), CB-RWU is not recommended; when the total runoff coefficient is high (Ψ ≥ 0.1), the reservoir volume is recommended to be 160 times the FAR or 3 times the value of non-potable water demand. With the aim of avoiding capacity waste, the amount of harvested rainwater should be maximized to meet the daily needs of residents.

Some related studies also showed the influence of these key factors on the environmental impacts of rainwater utilization. Santosh et al. showed that LCA tradeoffs exist due to variations in system requirements such as energy intensities and tank capacity for different rainfall and water demands [45]. In addition, the analysis of the optimal ratio between roof area and building height, i.e., the floor area ratio, would be useful to determine the best scenario of rainwater utilization [67]. Therefore, an optimization integrating the economic and water resource analysis of the system was necessary to evaluate the most cost-efficient option. A preliminary evaluation on the major types and trends of the environmental impacts was performed based on the characterization results of the LCA. However, due to the different units and calculation models of the indicators, direct comparisons were difficult. To overcome this limitation, the standardization factors from CML2001 were adopted for all impact indicators (Appendix A and Figure 5). After the standardization, the ADPF was observed to have the largest environmental impact, followed by the MAETP. This result agreed with some previous research. ADPF, MAETP and GWP of rainwater utilization had the largest impacts in each category, resources depletion, ecosystems and human health, respectively [23]. This is attributed to the large consumption of fossil energy, such as coal, oil, and natural gas, throughout all the life cycle stages, including raw material production, material transportation, on-site construction and end disposal, as well as the hard coal consumption used for the generation of electricity during the O&M phase. The distribution of eutrophication potential was the most discrete among all indicators due to changes in the concentration of pollutants (e.g., nitrogen and phosphorus) influenced by the total runoff coefficient during CB-RWU. In addition, it was recommended to not use polluting materials for roofs (e.g., linoleum roof materials) when building a new community or renovating an old community to reduce the environmental impact of CB-RWU during the whole life cycle.

ADPF contributed more than 99% of the total environmental impact, which could be explained by the consumption of fossil energy, such as coal, oil, and natural gas throughout all the life cycle stages. Based on the standardization results, the ADPF was selected as the main indicator for the environmental impacts of CB-RWU in the optimization model. The relationship between the environmental impact and three parameters, i.e., FAR, total runoff coefficient and reservoir volume, was built based on the ADPF from 80 scenarios. Equation (11) describes how the ADPF was calculated, and Appendix A presents the corresponding residual plot. The fitted data were evenly distributed on both sides of the original data, with a numerical error less than 25%. The R^2^ and the root mean squared error RMSE were equal to 0.8751 and 19.967, respectively, indicating the excellent fitting effect.
(12)$$ADP\;fossil=20.731FAR-246.091\psi +0.036S_{C}+102.996$$

### 3.2. Multi-Objective Optimization of CB-RWU

Table 4 reports the optimization results of the three model levels. Although both the economic and water resource benefits declined, the final result remained above 90% of the optimal value after the third-level optimization, with environmental impacts decreasing by 16%. This indicates the ability of the lexicographic method to solve the multi-objective optimization problem. Table 4 reveals that when the reservoir volume is about three times the user demand, the economic and water resource benefits of CB-RWU are high; when the reservoir volume is about two times the user demand, the comprehensive benefit is high. Table 5 presents the construction scheme of different underlying surfaces under the optimal solution. Comparing the FAR value, the area proportion of each underlying surface and the reservoir volume reveals that communities with low FAR values are candidates for CB-RWH and utilization in order to increase the comprehensive benefits. Moreover, the FAR and total runoff coefficient should be considered comprehensively to decide the suitable volume of reservoirs. These conclusions are consistent with the characterization results of LCA in Section 3.1.

The results relating to economic benefits in this study agree with those of previous research. Based on a case study consisting of a neighborhood of dense social housing (600 inhabitants/ha) with multi-storey buildings in Spain, Farreny et al., demonstrated that not all strategies were considered cost-efficient and that RWH strategies in dense urban areas were economically advantageous only if carried out at the appropriate scale in order to enable economies of scale [68]. Islam et al., calculated the net present value of rainwater utilization in a community in Bangladesh and also determined the scale effect of economic feasibility [69]. Ako et al., estimated the cost of rainwater utilization in a community in Cameroon, concluding that the promotion of CB-RWU could only be achieved with the economic and policy support from governments in developing countries [70]. Note that due to the differences in the economic and social background of the study areas, the design parameters of RWH systems and the cost specifications, direct comparison may exhibit inaccuracies. Nevertheless, the aforementioned studies can still indirectly indicate the low economic feasibility of CB-RWU across the globe and demonstrate the urgent requirement of governmental support for its further promotion and development.

The optimization effectiveness of the lexicographic method was further investigated using the traditional optimization approach, during which the importance of the three objective functions was considered equal. A lexicographic optimization was also conducted with environmental impact prioritized above the water-saving efficiency and benefit–cost ratio (Table 4). The ideal point method was combined under the traditional optimization to determine the optimal solution among all Pareto solutions that met the constraints, with objective function ranges of 0.0786–0.2456, 5.41–19.86% and 21.56–147.60 MJ for BCR, WSE and environmental impact, respectively. The values of the three objective functions of the optimal solution obtained by traditional optimization were 0.2456, 19.84% and 21.56 MJ for BCR, WSE and ADPF, respectively. This approach can best weigh the three objectives among all Pareto solutions, and the solution satisfies the actual needs and decision-making preferences of CB-RWU. In contrast, the corresponding values under the optimal solution from the lexicographic method with BCR prioritized were 0.3098, 28.47% and 24.68 MJ, and 0.3029, 27.45% and 24.38 MJ for environmental impact prioritized, respectively.

Comparing the results of the traditional optimization and lexicographic optimization with economic benefits prioritized reveals that the latter improved the BCR and WSE by 26% and 43%, respectively, while the environmental impact only increased by 14%. The lexicographic method not only greatly improved the optimization of the primary objectives, but also guaranteed the optimization of the secondary objectives. This indicates its effectiveness in solving optimization models with differences in the objective function importance. In terms of policy formulation, compared with the lexicographic optimization, the traditional optimization suggested a very small reservoir volume, which dramatically decreased the economic and water resource benefits. A small reservoir volume cannot easily meet the non-potable water demand, and inevitably leads to a waste of resources and energy during the construction and O&M phase. Compared with prioritizing the economic benefits, the optimization with environmental impact prioritized suggested more economic subsidies from the government. However, the water resource benefits were reduced, and the environmental impact was almost unchanged. In summary, the optimization of the lexicographic method with economic benefits prioritized was more practical, with higher comprehensive benefits and more reasonable policy recommendations.

### 3.3. Uncertainty Analysis

Random sampling was used to extract random parameter values based on the probability distributions of the uncertainty parameters. A total of 1000 sets of uncertainty parameter combinations were obtained and integrated into the objective functions for calculation by a random sampling program. Figure 6 presents the histogram of the simulated values for the three objective functions. Similar to the probability distribution of reservoir volume, i.e., left-skewed distribution, the probability of BCR ranging within 0.20–0.33 reached 70% (Figure 6a). In addition, in agreement with the probability distribution of the FAR and total runoff coefficient, i.e., left normal distribution, the probability of WSE ranging within 15–30% reached 60% (Figure 6b). Following the shape of a right-skewed distribution, the probability of an environmental impact lower than 150 MJ was 68% (Figure 6c). Comparing the three histograms reveals the BCR distribution to be most concentrated, followed by WSE and ADPF. The uncertainties of the three objective functions increased sequentially. Moreover, their data distributions were inconsistent with all uncertainty parameters, indicating the non-linear relationship between the model inputs and outputs.

In order to evaluate the representativeness of the simulation results, the average of the FAR, total runoff coefficient and reservoir volume were used to calculate the three objective functions. The results were then compared with those of the Monte Carlo simulation (Table 6). The coefficient of variation (C_V_) was used to describe the degree of dispersion of a set of data. The greater the degree of dispersion, the higher the uncertainty of the simulated objective function [71]. The C_V_ of the BCR was the lowest (0.395) among the three objective functions, while that of the ADPF was the highest (0.633). This is consistent with the trend shown in Figure 6. Thus, the uncertainty of the environmental impact was the highest with variations in the FAR, total runoff coefficient and reservoir volume, followed by WSE and BCR.

The simulation results obtained with the input of the average of each parameter were all lower than the average of the Monte Carlo simulation, indicating that the use of the average of each parameter underestimates (overestimates) the economic and water resource benefits (environmental benefits). In particular, the ADPF exhibited the largest deviation between the input of the averages and the Monte Carlo simulation results (22.2%), followed by the WSE (9.8%) and BCR (8.8%). The deviation sequence was the same as the C_V_ obtained from the Monte Carlo simulation (ADPF > WSE > BCR), indicating that the higher the uncertainty of the objective function, the larger the underestimation/overestimation deviation of the benefits.

In order to analyze the contribution of each parameter to the uncertainty of every objective function, the Monte Carlo simulation was performed on the random sampling of the FAR, total runoff coefficient and reservoir volume. This approach is distinct to the overall uncertainty analysis above (Table 7). The C_V_ of the BCR was consistently lower than the other two objective functions, indicating that it exhibited the lowest uncertainty from changes in the three parameters. The changes in the FAR and reservoir volume had a minimal impact on the uncertainty of the WSE and environmental impact, while changes in the total runoff coefficient had a far greater impact on the environmental benefits compared to the water resource benefits. Quantitative analysis of the uncertainty sources was then conducted (Figure 7). The uncertainty of the three objective functions mainly originated from changes in the total runoff coefficient, with a relative contribution rate of 80.0%, 68.5% and 84.4% for BCR, WSE and ADPF, respectively. Changes in the FAR and total runoff coefficient had the greatest impact on the uncertainty of environmental impact, while changes in the reservoir volume had the greatest impact on the WSE.

## 4. Conclusions

By taking the CB-RWU as the research object, this study integrated LCA with a multi-objective optimization model to optimize the relationship among all stages of CB-RWU including rainwater harvesting, storage, and distribution, considering the trade-offs among the BCR, WSE, and ADPF. The lexicographic method and Monte Carlo simulations were used for the multi-objective optimization and uncertainty analysis, respectively. The LCA results revealed the ADPF and marine aquatic ecotoxicity potential as the key life cycle impact indicators of CB-RWU. Based on the standardization results, the ADPF was selected as the representative environmental indicator for CB-RWU in the optimization model. The optimal solution from the lexicographic method was 0.3098, 28.47% and 24.68 MJ for the BCR, WSE and ADPF, respectively. Compared with the traditional optimization method, the lexicographic method improved the BCR by 26% and the WSE by 43%, while the ADPF increased by just 14%. In order to increase the comprehensive benefits, economic support from the government is urgently required for the further promotion and development of CB-RWU. The uncertainty of the ADPF was observed to be the highest, followed by the WSE and BCR. Changes in the total runoff coefficient were the main source of the uncertainty, suggesting that more attention should be paid to the area ratio of each underlying surface.

## Figures and Tables

**Figure 1 ijerph-20-02183-f001:**
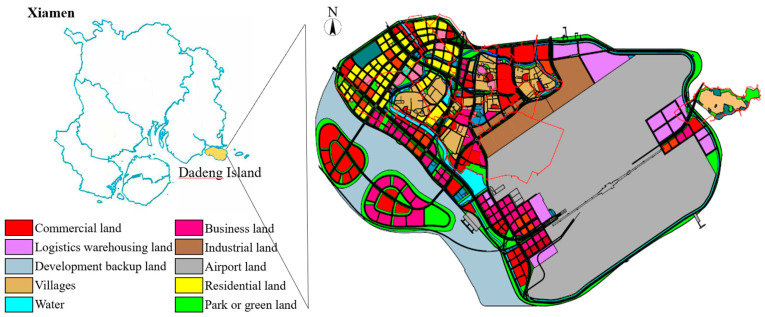
The location and land-use type of the study area.

**Figure 2 ijerph-20-02183-f002:**
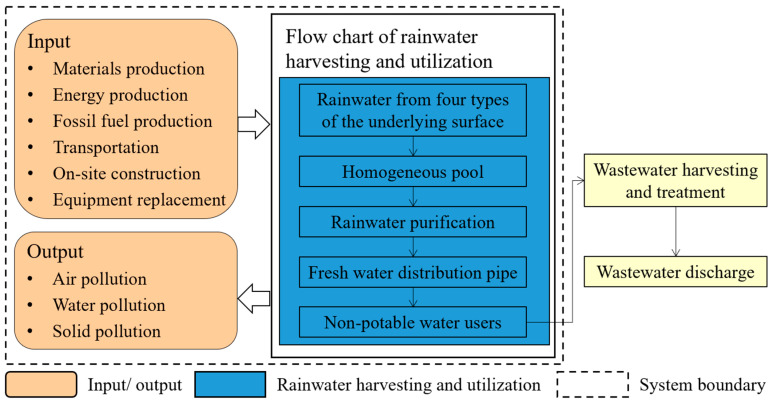
System boundary of rainwater harvesting and utilization for the life cycle assessment study.

**Figure 3 ijerph-20-02183-f003:**
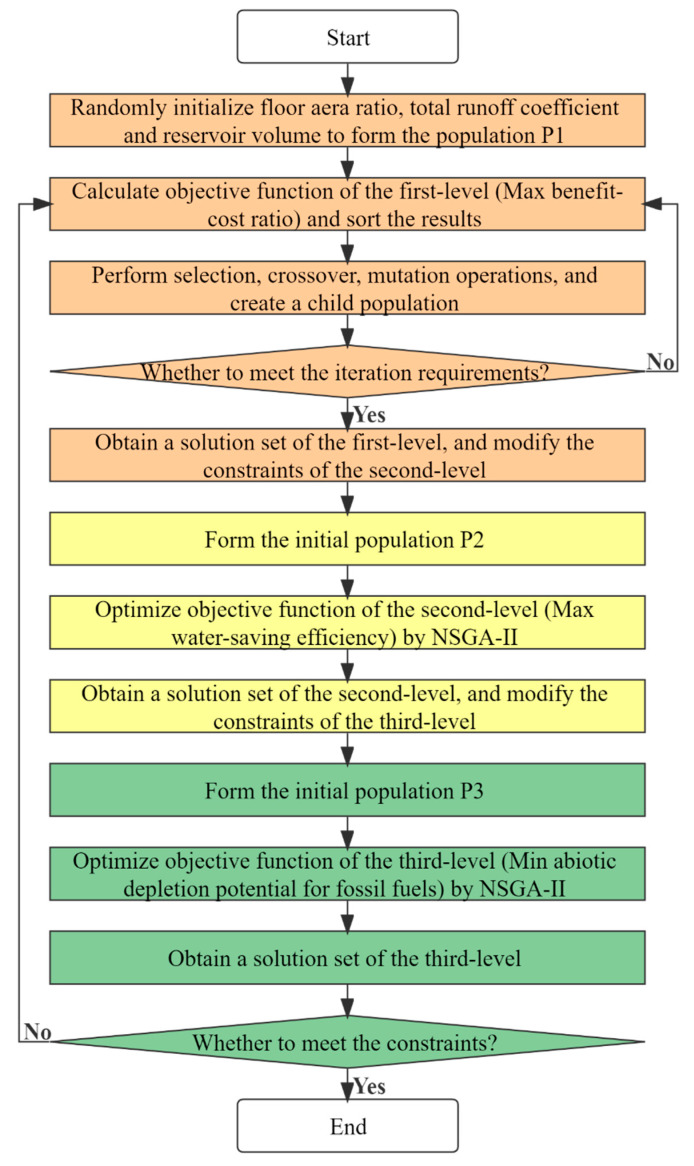
Methodology flowchart for multi-objective optimization of rainwater harvesting and utilization.

**Figure 4 ijerph-20-02183-f004:**
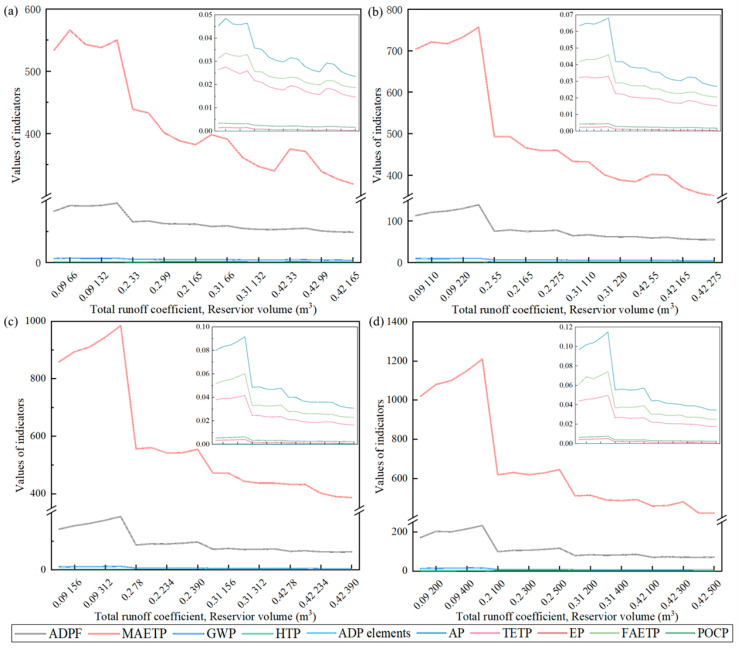
Life cycle assessment results of community-based rainwater utilization when the floor area ratio was: (**a**) 1; (**b**) 1.7; (**c**) 2.4; (**d**) 3.1. (EP: eutrophication potential; FAETP: freshwater aquatic ecotoxicity potential; POCP: photochemical ozone creation potential; ADP elements: abiotic depletion potential elements; AP: acidification potential; TETP: terrestrial ecotoxicity potential: global warming potential; HTP: human toxicity potential; ADPF: abiotic depletion potential for fossil fuels; MAETP: marine aquatic ecotoxicity potential).

**Figure 5 ijerph-20-02183-f005:**
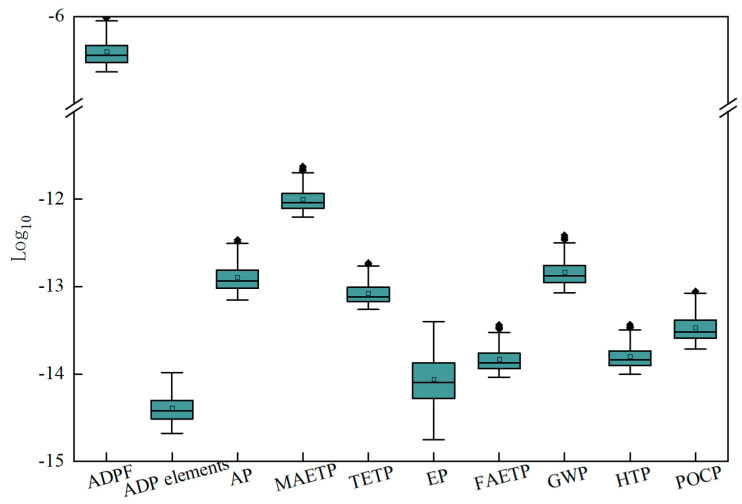
Normalization results of community-based rainwater utilization. (EP: eutrophication potential; FAETP: freshwater aquatic ecotoxicity potential; POCP: photochemical ozone creation potential; ADP elements: abiotic depletion potential elements; AP: acidification potential; TETP: terrestrial ecotoxicity potential: global warming potential; HTP: human toxicity potential; ADPF: abiotic depletion potential for fossil fuels; MAETP: marine aquatic ecotoxicity potential).

**Figure 6 ijerph-20-02183-f006:**
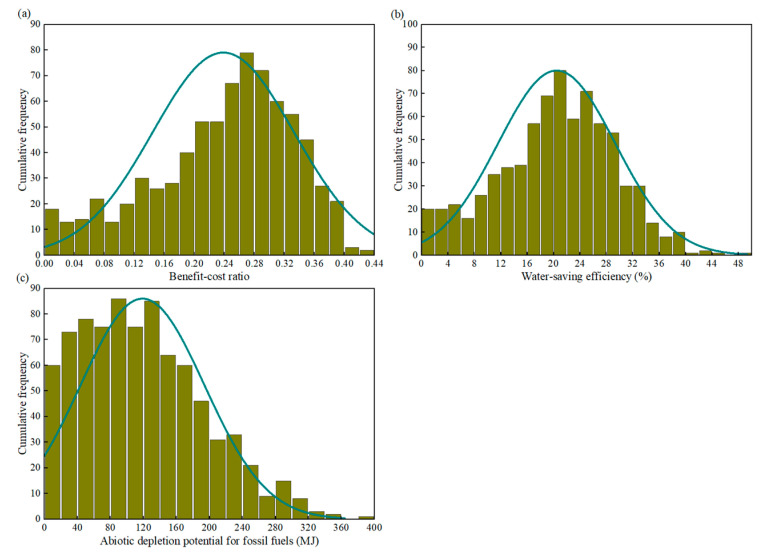
Simulated values histogram of: (**a**) benefit–cost ratio; (**b**) water-saving efficiency; (**c**) abiotic depletion potential for fossil fuels.

**Figure 7 ijerph-20-02183-f007:**
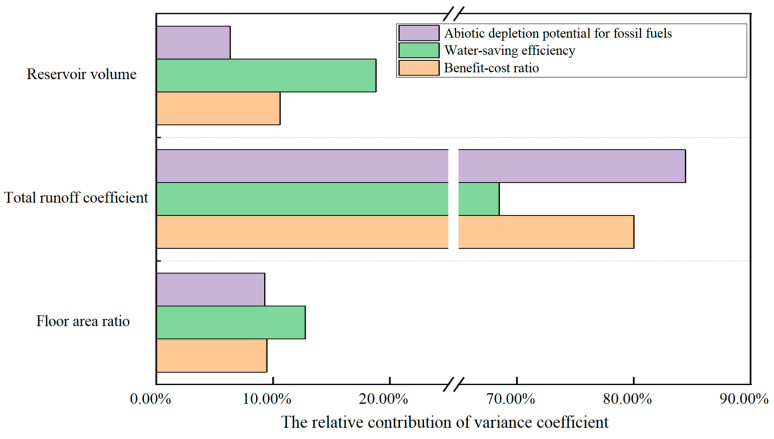
Analysis of source of uncertainty for objective functions.

**Table 1 ijerph-20-02183-t001:** Indicators of life cycle impact assessment.

Category	Indicators	Abbreviation	Unit
Resource depletion	Abiotic Depletion Potential for fossil fuels	ADPF	MJ
	Abiotic Depletion Potential elements	ADP elements	kg Sb eq.
Ecosystem	Acidification Potential	AP	kg SO_2_ eq.
	Eutrophication Potential	EP	kg PO_4_^3−^ eq.
	Marine Aquatic Ecotoxicity Potential	MAETP	kg DCB eq.
	Freshwater Aquatic Ecotoxicity Potential	FAETP	kg DCB eq.
	Terrestrial Ecotoxicity Potential	TETP	kg DCB eq.
Human health	Global Warming Potential	GWP	kg CO_2_ eq.
	Human Toxicity Potential	HTP	kg DCB eq.
	Photochemical Oxidant Creation Potential	POCP	kg Ethene eq.

**Table 2 ijerph-20-02183-t002:** Runoff coefficient for each underlying surface.

Number	Underlying Surface Type	Runoff Coefficient (Ψ_k_)
1	Green lands	0.15
2	Permeable pavement	0.29–0.36
3	Hard pavement	0.80–0.90
4	Roof	0.60–0.70

**Table 3 ijerph-20-02183-t003:** Values and probability distribution of model parameters.

Parameters	Probability Distribution	Range	Mathematical Expectation
Total runoff coefficient	Normal distribution	[0.09, 0.42]	0.255
Reservoir volume (m^3^)	Minimum distribution	[33, 500]	270.6
Floor area ratio (FAR)	Normal distribution	[1.0, 3.1]	2.05

**Table 4 ijerph-20-02183-t004:** Benefits analysis of community-based rainwater utilization scheme at each level.

	Benefit–Cost Ratio	Water-Saving Efficiency	Abiotic Depletion Potential for Fossil Fuels (MJ)	Floor Area Ratio	Total Runoff Coefficient	Reservoir Volume (m^3^)
**Lexicographic optimization with benefit–cost ratio prioritized above water-saving efficiency and environmental impact**						
The first-level	0.3347	32.71% *	26.31 *	1.000	0.420	165
The second-level	0.3288	31.63%	29.24 *	1.070	0.415	172
The third-level	0.3098	28.47%	24.68	1.017	0.420	124
**Lexicographic optimization with environmental impact prioritized above water-saving efficiency and benefit–cost ratio**						
The first-level	0.2456	19.84%	21.56	1.000	0.420	33
The second-level	0.3091	28.36%	24.63	1.000	0.420	118
The third-level	0.3029	27.45%	24.38	1.003	0.420	110
**Traditional multi-objective optimization**						
/	0.2456	19.84%	21.56	1.000	0.420	33

* The numbers were calculated from the optimal solution of this level.

**Table 5 ijerph-20-02183-t005:** The construction scheme of different underlying surfaces under the optimal solution.

Number	Underlying Surface Type	Runoff Coefficient (Ψ_k_)	Proportion of Area (β_k_)
1	green lands	0.15	35%
2	Permeable pavement	0.36	30%
3	Hard pavement	0.90	10%
4	Roof	0.70	25%

**Table 6 ijerph-20-02183-t006:** Assessment of Monte Carlo simulation results.

Objective Functions	Average Input ^1^	Average ^2^	Standard Deviation ^2^	Coefficient of Variation (C_V_) ^2^
Benefit-cost ratio	0.217	0.238	0.094	0.395
Water-saving efficiency	18.5%	20.5%	0.089	0.433
Abiotic depletion potential for fossil fuels	92.5 MJ	118.8 MJ	75.222 MJ	0.633

^1^ The results were calculated by using the average of each parameter. ^2^ The results were calculated by the Monte Carlo simulation.

**Table 7 ijerph-20-02183-t007:** Assessment of Monte Carlo simulation results.

Coefficient of Variation (C_V_)	Benefit–Cost Ratio (BCR)	Water-Saving Efficiency (WSE)	Abiotic Depletion Potential for Fossil Fuels (ADPF)
Floor-area ratio (FAR)	0.078	0.124	0.178
Total runoff coefficient	0.661	0.664	1.620
Reservoir volume	0.087	0.182	0.121

## Data Availability

The data presented in this study are available in the article or Appendix A.

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
