# Peer review of "The Life cycle Assessment Integrated with the Lexicographic Method for the Multi-Objective Optimization of Community-Based Rainwater Utilization"

_ijerph, 2023, doi:10.3390/ijerph20032183_

Round 1

Reviewer 1 Report

manuscript title: “The life cycle assessment integrated with the  lexicographic method for the multi-objective optimization of community-based rainwater utilization-ID=ijerph-2116116”.

 The main queries which need to reply are listed below:

 1- I recommend for selecting a better title for the manuscript.

2- Abstract is not prepared in a scientific structure and needs to be modified. Maybe, adding some quantified results helps to improve it.

3- Study area and other useful information should be shown in a Figure. What do you mean by “Figure A1” in line 140?

4- Adding some descriptions about rainwater management in the study area is necessary.

5- Line 185, which 30 years were used. Please clarify about the source of input data in Figure 1.

6- The amounts of last column in Table 2 (runoff coefficients) are obtained for your study or are for literature review? If the second is true, please state the related references.

7- Table A3 (line 356) is not shown in the manuscript.

8- What is the conceptual (physical) meaning of the indicators which are used in Figure 3? Also this Figure is not clear and needs to be modified.

9- Results, both Figures and Tables, should be discussed more in details.

10- An important point regarding Results and Discussion is that no comparisons are made between the results of this research and those of similar works. I can also suggest that the author(s) invest their efforts in discussing the results and their possible reasons. This will strengthen the scientific vigour of the manuscript.

11- Finally, how do you validate your proposed results?

Reviewer 2 Report

This study integrated the life cycle assessment (LCA) with a multi-objective optimization model to optimize the relationship among all stages of CB-RWU, considering the trade-offs among the benefit-cost ratio, water-saving efficiency, and environmental impact. The optimal solution was obtained through the nondominated sorting genetic algorithm II and lexicographic method, with Monte Carlo simulation used for uncertainty analysis. In general, this topic is interesting and the manuscript is well presented. There is only some minor parts need to be revised.

(1) Some references should be added e.g. L366-368

(2) Figure 3 MAETP blue line can be revised to make it more clear.

(3) The abbreviation should also be explained in Figure 4 caption.

Reviewer 3 Report

Great job.

1. Suplementary materials - very clear in text and in message. 

2. Title, abstract, key words - clear and good

3. Glossary - The most important thing - it is at the beginning!

4. Intro - great.

5. Line 335: [55]:1 <-- provide space between ] and : 

6. Equation (11) - formatting to the left

Reviewer 4 Report

The paper appears good and interesting but I think the following comments must be considered to enrich the text.

- Details of the objectives should be presented at end of the INTRODUCTION.

- The map of the study area should be presented. The following reference can help you:

“Aquifer-wide estimation of longitudinal dispersivity by the combination of empirical equations, inverse solution, and aquifer zoning methods”

“Future Runoff Variation and Flood Disaster Prediction of the Yellow River Basin Based on CA-Markov and SWAT”

- More descriptions of optimization can be presented. The following reference can help you:

“Baipenzhu Reservoir Inflow Flood Forecasting Based on a Distributed Hydrological Model”

- It is better to present more numerical results in ABSTRACT

- The hydrological characteristics of study sites should be presented. The following reference can help you:

“Hydrological modelling through SWAT over a Himalayan catchment using high-resolution geospatial inputs”

- The fitted pdf for Figure 5-a is not suitable. Can you present more description?

- The investigation of the non-stationary condition can enrich your research. This paper may be interesting from a dynamic behavior point of view:

“Estimation of non-stationary behavior in annual and seasonal surface freshwater volume discharged into the Gorgan Bay, Iran”

“Statistical analysis of attributions of climatic characteristics to nonstationary rainfall‐streamflow relationship”

Round 2

Reviewer 1 Report

Daer author(s)

Thanks for your attempt to answer the queries.

Best regards